# Modeling Balanced Explicit and Implicit Relations with Contrastive Learning for Knowledge Concept Recommendation in MOOCs

## ABSTRACT

The knowledge concept recommendation in Massive Open Online Courses (MOOCs) is a significant issue that has garnered widespread attention. Existing methods primarily rely on the explicit interactions between users and knowledge concepts on the MOOC platforms for recommendation. However, there are numerous implicit relations (e.g., shared interests or same knowledge levels between users) generated within the users' learning activities on the MOOC platforms. Existing methods fail to consider these implicit relations, and these relations themselves are difficult to learn and represent, causing poor performance in knowledge concept recommendation and an inability to meet users' personalized needs. To address this issue, we propose a novel framework based on contrastive learning, which can represent and balance the explicit and implicit relations for knowledge concept recommendation in MOOCs (**CL-KCRec**). Specifically, we first construct a MOOCs heterogeneous information network (MHIN) by modeling the data from the MOOC platforms. Then, we utilize a relation-updated graph convolutional network (GCN) and stacked multi-channel graph neural network (GNN) to represent the explicit and implicit relations in the MHIN, respectively. Considering that the quantity of explicit relations is relatively fewer compared to implicit relations in MOOCs, we propose a contrastive learning with prototypical graph to enhance the representations of both relations to capture their fruitful inherent relational knowledge, which can guide the propagation of students' preferences within the MHIN. Based on these enhanced representations, to ensure the balanced contribution of both towards the final recommendation, we propose a dual-head attention mechanism for balanced fusion. Experimental results demonstrate that CL-KCRec outperforms several state-of-the-art baselines on real-world datasets in terms of HR, NDCG and MRR.

## CCS CONCEPTS

• **Applied computing** → *E-learning*; • **Information systems** → **Recommender systems**; *Personalization*; • **Computing methodologies** → **Neural networks**.

## KEYWORDS

Knowledge concept recommendation, Explicit and implicit Relations, Contrastive learning, Attention mechanism

**ACM Reference Format:**
Anonymous Author(s). 2024. Modeling Balanced Explicit and Implicit Relations with Contrastive Learning for Knowledge Concept Recommendation in MOOCs. In *Proceedings of the ACM Web Conference (WWW'2024)*. ACM, New York, NY, USA, 9 pages. https://doi.org/XXXXXXX.XXXXXXX

## 1 INTRODUCTION

In recent years, the online learning industry has experienced rapid growth [19], which is becoming an integral part of the modern education system. Among these, massive open online courses (MOOCs), as a representative of this transformation, are becoming a popular educational mode worldwide[8]. Although the number of new users on the various MOOC platforms continues to rise, a primary problem remains the low course completion rate[36]. Many users struggle to complete all the knowledge concepts in a course, which leads to inefficiency or even dropout. For example, a course about triangles might cover various knowledge concepts such as sides, angles, auxiliary lines, the Pythagorean theorem, etc. Hence, it is crucial to capture personalized user interests and then recommend specific knowledge concepts in MOOCs.

Existing methods for knowledge concept recommendation in MOOCs utilize the explicit relations among users, knowledge concepts, etc. [8] proposed an end-to-end graph neural network based approach with attention mechanism to capture the various relations in MOOCs. However, there are not only explicit relations in users' learning activities, but also numerous implicit relations like latent social connections, shared interests, similar knowledge levels, etc. As shown in Figure 1, besides observing explicit relations, such as users learning courses and watching videos, we can also discover many inherent implicit relations. For instance, if both user_1 and user_2 choose to learn the same course_1, they might share common interests. Furthermore, if user_1 watched video_1 but clicked only on knowledge_concept_1, then knowledge_concept_2 might be his next focus or something he needs to supplement. Ignoring these implicit relations will undoubtedly impact the effectiveness of knowledge concept recommendation.

Consequently, it is a significant challenge to thoroughly represent and leverage implicit relations in enhancing the knowledge concept recommendation, as illustrated by the following three challenges. First, a user's history in MOOCs mainly describes explicit relations; it is crucial to automatically represent the inherent implicit relations based on these explicit relations. Second, explicit relations are obviously fewer in number than implicit relations, while implicit relations are also more complex than explicit ones during the actual learning process, which brings the challenge of effectively guiding the propagation of students' preferences. It is

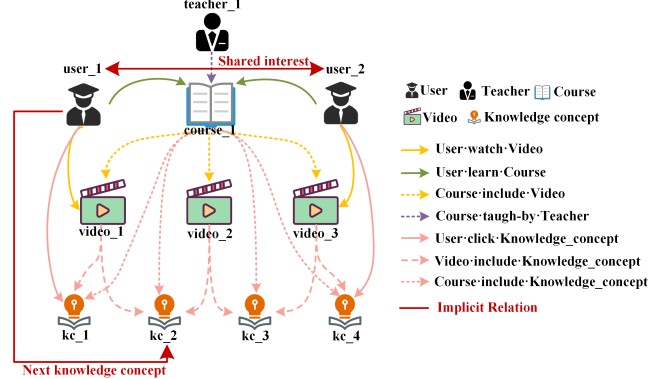

**Figure 1: A comprehensive view of data in MOOCs.**

necessary to enhance the representations of both types of relations. This will capture their inherent relational knowledge, which can be utilized to guide the propagation of students' preferences. Third, it is essential to balance the contributions of the explicit and implicit relations in the knowledge concept recommendation task.

To address this issue, we propose a novel framework based on contrastive learning, design to represent and balance the explicit and implicit relations for knowledge concept recommendation in MOOCs (**CL-KCRec**). First, we construct a MOOCs heterogenous information network (MHIN) using data from the MOOC platforms. Then, we propose an explicit relation learning module based on relation-updated GCN, and an implicit relation learning module based on a stacked multi-channel GNN, which represents multi-hop relations through a soft attention selection mechanism. We also propose a contrastive learning with prototypical graph to enhance the representations of both relations, and propose a dual-head attention mechanism for balanced fusion. Experimental results demonstrate that CL-KCRec outperforms several state-of-the-art baselines on real-world datasets in terms of HR, NDCG and MRR.

## 2 RELATED WORK

This work is mainly relevant to knowledge concept recommendation in MOOCs and contrastive learning for recommender systems.

### 2.1 Knowledge Concept Recommendation

Knowledge concept recommendation is an essential component of personalized learning in MOOCs. Existing methods can be primarily categorized into three types: collaborative filtering (CF)-based methods [22, 27], heterogeneous information network (HIN)-based methods[8, 28] and reinforcement learning (RL)-based methods[7, 11, 17]. CF-based methods, which take into account users' historical interactions, have achieved success in traditional recommendation strategies. [22] introduce the core concepts of collaborative filtering and design rating systems for recommendation. HIN-based methods incorporate users' historical interactions into a heterogeneous information network and optimize the representations for recommendation. [8] is a state-of-the-art method that employs an

attention-based graph convolutional network, which utilizes meta-paths to obtain the representation of nodes for knowledge concept recommendation in MOOCs. RL-based methods apply reinforcement learning for recommendation to adaptively update the strategy during long-term interaction. [7] propose the reinforced strategy that can recommend the items with substantial long-term benefits.

### 2.2 Contrastive Learning for Recommendation

Recently, contrastive learning has received widespread attention for its ability to provide powerful self-supervised signals in various fields, such as natural language processing[5, 16] and computer vision[3]. By contrasting positive and negative samples from different views, contrasting learning can learn high-quality and discriminative representations, ensuring sample balance within specific scenarios or tasks. Some studies have attempted to apply the contrastive learning approach to recommendation tasks[18, 29, 30]. [31] proposed a contrastive learning framework for KG-enhanced recommendation.[1] proposed heterogeneous information network Contrastive Learning. To adapt the contrastive learning into our work, we propose a novel contrastive learning approach based on prototypical graphs to enhance the representations of users and knowledge concepts for the knowledge concept recommendation in MOOCs.

## 3 PRELIMINARIES

In this section, we introduce the definitions involved in our work.

**Task Description.** Given a target user with corresponding interactive data in MOOCs, the goal is to calculate the user's interest score for a series of knowledge concepts and generate a recommended list of the top $N$ knowledge concepts. More formally, given the interactive data of a user, denoted as $u_i$, a predict function $f$ is learned and utilized to generate a recommendation list of knowledge concepts, where each concept is denoted as $k_j$, for $f : u_i \to \{k_j\}_{j=1}^{N}$.

**Definition 1: MOOCs Heterogeneous information network (MHIN).** In this work, we denote the MHIN as $\mathcal{G} = (\mathcal{V}, \mathcal{E})$, consisting of the node set $\mathcal{V}$ and the edge set $\mathcal{E}$. Each node $v_i \in \mathcal{V}$ is associated with a node type mapping function $f_v : \mathcal{V} \to \mathcal{T}^v$ and each edge is associated with an edge type mapping function $f_e : \mathcal{E} \to \mathcal{T}^e$. The MHIN can be represented by a collection of adjacency matrices $\mathbb{A} = \{A_t\}_{t=1}^{|\mathcal{T}^e|}$, where $A_t \in \mathbb{R}^{|\mathcal{V}| \times |\mathcal{V}|}$ denotes an adjacency matrix where $A_t[i, j]$ is non-zero if there exists a $t$-th type edge from node $v_j$ to node $v_i$.

**Definition 2: Explicit Relations in MHIN (ER).** We define the edges between a specific entity and all of its single-hop neighbor nodes as the explicit relations of that entity. As shown in Figure 2(a), the user_1 has various explicit relations with the knowledge concept_1, the course_1, and the video_1, respectively.

**Definition 3: Implicit Relations in MHIN (IR).** We define the implicit relations as the complex multi-hop relations involving multiple entities and their associated explicit relations. As shown in Figure 2(c), both user_1 and user_2 clicked on knowledge_concept_1. This suggests that they might share a common interest, which could be valuable for knowledge concept recommendation. Obviously, some of these implicit relations are simple and interpretable, while

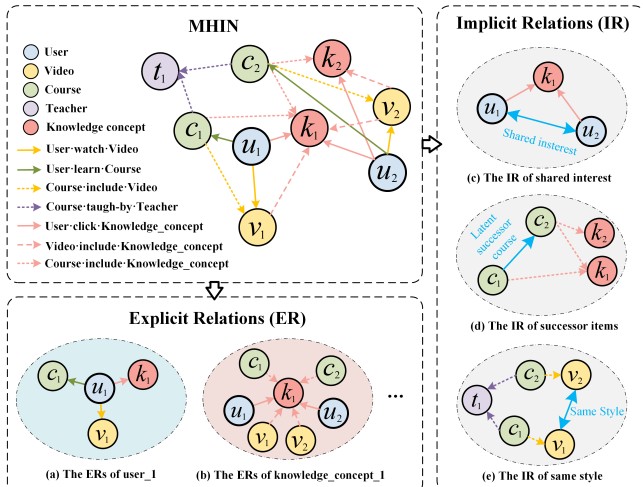

**Figure 2: The Explicit and Implicit Relations in MHIN.**

others are complex and harder to explain but remain crucial for recommendation. The implicit relation learning method we propose could effectively address this issue.

## 4 CL-KCREC

The architecture of our proposed knowledge concept recommendation framework, **CL-KCRec**, is shown in Figure 3. Each component module will be presented in detail in the following sections.

### 4.1 Construction of MHIN

We model the data from platforms as a MHIN (shown in Figure 3(b)) that consists of five types of nodes: users (**U**), knowledge concepts (**K**), courses (**C**), videos (**V**), and teachers (**T**). These nodes are connected via seven types of edges, which are represented as the following adjacency matrices:

- $\mathbf{A}^{UK}$: *user·click·knowledge_concept* matrix, where each element indicates whether a user clicked a knowledge concept.
- $\mathbf{A}^{UV}$: *user·watch·video* matrix, where each element indicates whether a user watched a video.
- $\mathbf{A}^{UC}$: *user·learn·course* matrix, where each element indicates whether a user learned a course.
- $\mathbf{A}^{VK}$: *video·include·knowledge_concept* matrix, where each element indicates whether a knowledge concept is included in a video.
- $\mathbf{A}^{CK}$: *course·include·knowledge_concept* matrix, where each element indicates whether a knowledge concept is involved in a course.
- $\mathbf{A}^{CV}$: *course·include·video* matrix, where each element indicates whether a course includes a video.
- $\mathbf{A}^{CT}$: *course·taught_by·teacher* matrix, where each element indicates whether a course is taught by a teacher.

After constructing the MHIN, we feed it into subsequent modules to represent both explicit and implicit relations, respectively.

## 4.2 Explicit Relation Learning

In this section, we leverage the Knowledge Graph Embedding techniques to jointly embed nodes and their explicit relations within MHIN, as shown in Figure 3(c).

*4.2.1 Multi-Relational Representations.* . To learn the representation in MHIN with various explicit relations, we first need to represent these relations. To alleviate the over-parameterization issue in graph representation learning, we draw inspiration from the variant of the basic decomposition approach[23, 25]. We use a linear combination of a set of basis vectors to represent each explicit relation instead of defining a separate embedding vector. Hence, the initial representation of an explicit relation $r$ is given as:

$$z_r = \sum_{b=1}^{\mathcal{B}} \alpha_{br} c_b \tag{1}$$

where $z_r \in \mathbb{R}^{d_1}$ denotes the representation of $r$-th explicit relation. $c_b \in C$ denotes $b$-th basis vector where $C = \{c_1, c_2, \cdots, c_B\}$. $\alpha_{br}$ is a learnable scalar weight.

*4.2.2 Relation-Updated GCN.* . The original GCN update equation is given by:

$$h_{v_i} = f_{agg}\big(\sum_{(v_j, r) \in \mathcal{N}_{(v_i)}} W_r h_{v_j}\big) \tag{2}$$

where $\mathcal{N}_{(v_i)}$ represents the neighbor nodes that have explicit relations with $v_i$, and $W_r$ denotes the learnable parameters. To incorporate the explicit relation representation $z_r$ into GCN, the entity-relation composition operation[25] is used, which is given as:

$$x_{v_i} = \varphi(x_{v_j}, z_r) \tag{3}$$

where $\varphi$ is a composition operator for which we adopt the non-parameterized operation of circular-correlation as proposed by[20]. $v_j, r$, and $v_i$ denote the head node, explicit relation and tail node. $x_{v_i}, x_{v_j} \in \mathbb{R}^{d_0}$ denote the initial representation of nodes by BERT[13] with their auxiliary information. The equation of the relational-updated GCN is given as:

$$h_{v_i|er} = f_{agg}\big(\sum_{(v_j, r) \in \mathcal{N}_{(v_i)}} W_r \varphi(x_{v_j}, z_r)\big) \tag{4}$$

where $h_{v_i|er}$ denotes the representation of node $v_i$ updated by explicit relations. The representation $z_r$ is also transformed as follows:

$$h_r = W_{rel} z_r \tag{5}$$

where $W_{rel} \in \mathbb{R}^{d_1 \times d_0}$ represents a learnable transformation matrix. Consequently, we extend Eq.(4) to the $l$ layers. Let $h_{v_i|er}^{(l)}$ denote the final representation of node $v_i$, which is given as:

$$h_{v_i|er}^{(l)} = f_{agg}\big(\sum_{(v_j, r) \in \mathcal{N}_{(v_i)}} W_r^{(l-1)} \varphi(h_{v_j|er}^{(l-1)}, h_r^{(l-1)})\big) \tag{6}$$

Similarly, let $h_r^{(l)} = W_{rel}^{(l-1)} h_r^{(l-1)}$ denote the representation of the explicit relation $r$ after $l$ layers, $h_{v_i|er}^{(0)}$ and $h_r^{(0)}$ respectively correspond to the initial representations of the node $x_{v_i}$ and the explicit relation $z_r$.

### 4.3 Implicit Relationship Learning

In this section, we propose a stacked multi-channel GNN to represent implicit relations in MHIN, as shown in Figure 3(d).

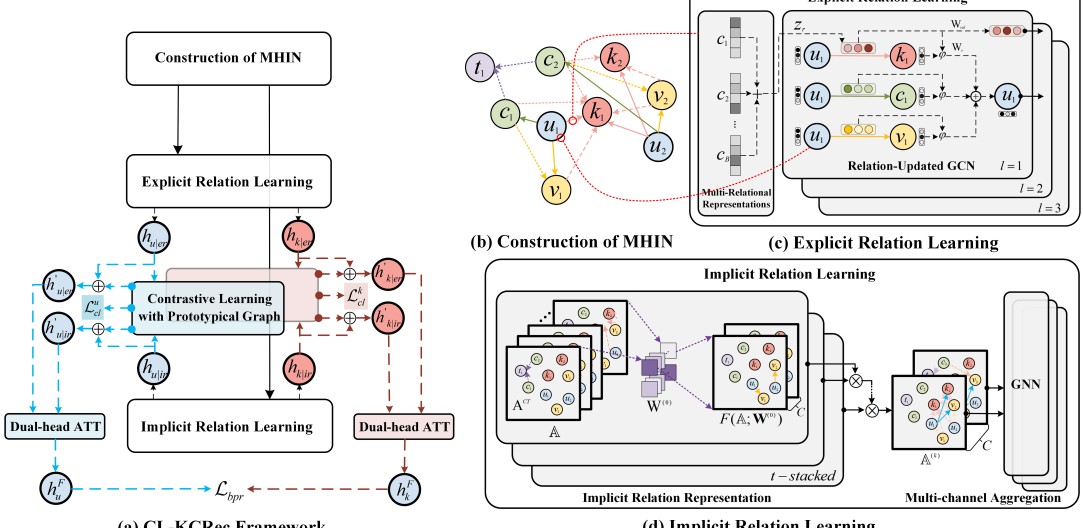

**Figure 3: The overall architecture of CL-KCRec.**

*4.3.1 Implicit Relation Representation.* . Each $\mathbf{A} \in \mathbb{A}$ represents a graph structure corresponding to a specific explicit relation in MHIN. Inspired by [34], we use a soft attention selection mechanism to automatically select the new graph structure to represent the multi-hop relation, that is, the implicit relation. Specially, the $1 \times 1$ convolution with the weights from $softmax$ function is used as:

$$F(\mathbb{A}; W) = conv_{1 \times 1}(\mathbb{A}; softmax(W)) = \sum_{i=1}^{|\mathcal{T}^e|} \alpha_i A_i \quad (7)$$

where $W \in \mathbb{R}^{1 \times 1 \times |\mathcal{T}^e|}$ denotes the learnable parameter matrix. The soft selection from different explicit relations is realized by the convex combination of adjacency matrices as $\alpha \cdot \mathbb{A}$ [2]. Then, we stack this operation over $t$ layers to get the $t$ soft-selected adjacency matrices. By conducting matrix multiplication on them in a layer-sequential manner, we obtain a new adjacency matrix that represents the $t$-hops implicit relation as:

$$A^{(t)} = (D^{(t)})^{-1} A^{(t-1)} F(\mathbb{A}; W^{(t)}) \quad (8)$$

where $A^{(0)} = F(\mathbb{A}; W^{(0)})$ and $D^{(t)}$ represents a degree matrix to normalize $A^{(t)}$ to ensure numerical stability. Hence, we represent an implicit relation from MHIN with an arbitrary maximum length of $(t + 1)$-hops, which is expressed as a new graph structure and represented as the adjacency matrix $A_{ir}$ :

$$A_{ir} = \left( \sum_{i_0=1}^{|\mathcal{T}^e|} \alpha_{i_0}^{(0)} A_{i_0} \right) \cdot \left( \sum_{i_1=1}^{|\mathcal{T}^e|} \alpha_{i_1}^{(1)} A_{i_1} \right) \cdots \left( \sum_{i_t=1}^{|\mathcal{T}^e|} \alpha_{i_t}^{(t)} A_{i_t} \right) \quad (9)$$

*4.3.2 Multi-channel Aggregation.* . Considering that users in MOOCs may be influenced by multiple implicit relations simultaneously, we extend Eq.(8). The original equation can only represent a single

implicit relation at a time; we modify it to support the simultaneous generation of multiple implicit relations by setting the convolution filter channels to $C$:

$$\mathbb{A}^{(t)} = (\mathbb{D}^{(t)})^{-1} \mathbb{A}^{(t-1)} F(\mathbb{A}; W^{(t)}) \quad (10)$$

where $\mathbb{A}^{(t-1)} F(\mathbb{A}; W^{(t)}) = \|_{c=1}^{C} (A_c^{(t-1)} F(\mathbb{A}; W^{(t,c)})$. $C$ denotes the number of channels, and $\mathbb{D}^{(t)}$ represents a set of degree tensors. Each channel of the output tensor $\mathbb{A}^{(t)}$ is fed into $l$ GNN layers to update the representations of nodes:

$$h_{v_i|ir}^{(l)} = f_{agg} \left( \|_{c=1}^{C} \sigma(\hat{D}_c^{-1} \hat{A}_c^{(t)} h_{v_i|ir}^{(l-1)} W^{(l-1)}) \right) \quad (11)$$

where $h_{v_i|ir}^{(l)}$ denotes the final representation of node $v_i$ at the $l$-th GNN layer. $W^{(l)} \in \mathbb{R}^{d^{(l)} \times d^{(l+1)}}$ represents a learnable weight matrix. $h_{v_i|ir}^{(0)} = x_{v_i}$ denotes the initial representation.

## 4.4 Contrastive Learning with Prototypical Graph

As the issue has been described in Section 1, the quantity of explicit relations is obviously fewer than that of implicit relations. Hence, after obtaining the node representations with explicit and implicit relations, to maximize the effectiveness of these two representations in recommendation, we propose a contrastive learning with prototypical graph approach to enhance the representations aimed at capturing both explicit and implicit knowledge feature, which can guide the propagation of students' preferences. The architecture is shown in Figure 4. For convenience, we denote the *original representations* of the user nodes obtained as $h_{u|er}$ and $h_{u|ir}$, and the knowledge concept nodes as $h_{k|er}$ and $h_{k|ir}$.

*4.4.1 Prototypes Generation.* We perform clustering method to cluster the users' *original representations* with explicit relations $h_{u|er} = \{h_{u_i|er}\}_{i=1}^{N_u}$ (where $N_u$ denotes the number of users.) to generate

**Figure 4: The architecture of Contrastive Learning with Prototypical Graph Module.**

$n$ clusters as prototypes $C = \{c_i\}_{i=1}^n$, collectively representing the embedding space of explicit relations. Here, a prototype[14] is defined as *a representative embedding for a group of semantically similar instances*.

*4.4.2 Prototypical Graph Generation.* For the original representation $h_{u_i|er}$ of the user $u_i$ (where we set the sample number with the size of mini-batch at each training epoch, considering the limitation of computational cost and memory), we treat it and all the prototypes $C$ as nodes $\mathcal{V}_{u_i|er} = [h_{u_i|er}, c_1, c_2, \cdots, c_n]$ in the prototypical graph, and then fully connect these nodes $\mathcal{E}_{u_i|er}$ to get the adjacency matrix $A_{u_i|er} \in \mathbb{R}^{(n+1) \times (n+1)}$ of the prototypical graph. Ultimately, the prototypical graph can be represented as $\mathcal{G}_{u_i|ex} = (\mathcal{V}_{u_i|er}, \mathcal{E}_{u_i|er})$. Analogously, we can generate the prototypical graph $\mathcal{G}_{u_i|ir}$ with implicit relations, as well as $\mathcal{G}_{k_j|er}$ and $\mathcal{G}_{k_j|ir}$ for knowledge concept $k_j$.

*4.4.3 Graph Attention Networks.* The prototypical graph $\mathcal{G}_{u_i|er}$ of user $u_i$ is input into the graph attention network (GAT)[26]. Then, we obtain the *enhanced representation* $z_{u_i|er}$ of user $u_i$:

$$z_{u_i|er} = f(GAT(\mathcal{V}_{u_i|er}, \mathcal{E}_{u_i|er})) \tag{12}$$

*4.4.4 InfoNCE-based Contrastive Loss.* . We propose to model the original and enhanced representation with *InfoNCE*-based contrastive learning loss. Specifically, for explicit relations, the enhanced $z_{u|er}$ are considered as the *positive* samples for the *anchor* $h_{u|er}$, while the enhanced $z_{u|ir}$ are considered as the *negative* samples. Conversely, for implicit relations, the augmented $z_{u|ir}$ are considered as the *positive* samples for the *anchor* $h_{u|ir}$, and the augmented $z_{u|er}$ are considered as the *negative* samples, as follows:

$$\mathcal{L}_1 = \sum_{i=1}^{N_u} -\log \frac{exp\left(s(h_{u_i|er}, z_{u_i|er})/\tau\right)}{\sum_{j=1}^{N_u} exp\left(s(h_{u_i|er}, z_{u_j|ir})/\tau\right)}$$

$$\mathcal{L}_2 = \sum_{i=1}^{N_u} -\log \frac{exp\left(s(h_{u_i|ir}, z_{u_i|ir})/\tau\right)}{\sum_{j=1}^{N_u} exp\left(s(h_{u_i|ir}, z_{u_j|er})/\tau\right)} \tag{13}$$

$$\mathcal{L}_{cl}^u = \mathcal{L}_1 + \mathcal{L}_2$$

where $s(\cdot)$ represents the similarity function, which can be either an inner product or cosine similarity (we adopt the latter). $\tau$ represents the temperature coefficient. Analogously, we can derive the

loss $\mathcal{L}_{cl}^k$ for the knowledge concepts perspective. The combined contrastive loss is expressed as:

$$\mathcal{L}_{cl} = \alpha_u * \mathcal{L}_{cl}^u + \alpha_k * \mathcal{L}_{cl}^k \tag{14}$$

where $\alpha_u$ and $\alpha_k$ represent two hyperparameters.

## 4.5 Fusion and Optimization

We adopted a dual-head attention mechanism to fuse the enhanced representations, which are situated in separate vector spaces respective to their corresponding explicit or implicit relations by contrastive learning with prototypical graph, into a unified high dimensional vector space to balance their contributions for knowledge concept recommendation task.

*4.5.1 Dual-head Attention Fusion.* . we concatenate the optimized original and enhanced representations of users to get the output representations:

$$h'_{u|er} = h_{u|er} \oplus z_{u|er}$$
$$h'_{u|ir} = h_{u|ir} \oplus z_{u|ir} \tag{15}$$

where $\oplus$ denotes the concatenation operation. Similarly, we can obtain the output representations of knowledge concepts $h'_{k|er}$ and $h'_{k|ir}$. Afterward, we map the representations into the same vector space, then fuse them using cross-modal attention as follows:

$$\hat{h}_{u|er} = W_v tanh(W_{er} h'_{u|er} + b_{er}) + b_v$$
$$\hat{h}_{u|ir} = W_v tanh(W_{ir} h'_{u|ir} + b_{ir}) + b_v$$
$$\alpha_{u|er} = W_s tanh(W_{er} h'_{u|er} + b_{er}) + b_s$$
$$\alpha_{u|ir} = W_s tanh(W_{ir} h'_{u|ir} + b_{ir}) + b_s \tag{16}$$
$$\tilde{\alpha}_{u|er}, \tilde{\alpha}_{u|ir} = softmax(\alpha_{u|er}, \alpha_{u|ir})$$
$$h_u^F = \tilde{\alpha}_{u|er} \cdot \hat{h}_{u|er} + \tilde{\alpha}_{u|ir} \cdot \hat{h}_{u|ir}$$

where $h_u^F$ is the fused representation. We use shared weights $W_v$ and bias $b_v$ to map the representations from their respective vector space into a unified high-dimensional vector space, while using shared attention weights $W_s$ and biases $b_s$ to align the attention coefficients. Similarly, we can obtain the fused representation vector of the knowledge concepts $h_k^F$.

### 4.5.2 Optimization Objectives.

In our work, we employ the dot-product to forecast $\hat{y}_{u_i,k_j} = {h_{u_i}^F}^T \cdot h_{k_j}^F$ to forecast the interaction likelihood between user $u_i$ and knowledge concept $k_j$. $\hat{y}_{u_i,k_j} \in \mathbb{R}$ denotes the score that indicates the likelihood of user $u_i$ interacting with knowledge concept $k_j$. A larger value of $\hat{y}_{u_i,k_j}$ indicates a higher probability of interaction. We use the Bayesian Personalized Ranking (BPR) pairwise loss function [21]. Specifically, each training sample is prepared with a user $u_i$, a positive knowledge concept $k_j^+$ with which the user has interacted, and a negative knowledge point $k_l^-$ with which the user has not interacted. For each training sample, we maximize the prediction score as follows:

$$\mathcal{L}_{bpr} = \sum_{(u_i,k_j^+,k_l^-) \in O} -ln(\sigma(\hat{y}_{u_i,k_j^+} - \hat{y}_{u_i,k_l^-})) + \lambda\|\Theta\|^2 \qquad (17)$$

where $ln(\cdot)$ and $\sigma(\cdot)$ denote the logarithm function and the sigmoid function. $\lambda$ denotes the hyperparameter for the weight of the regularization term. Combining the BPR loss function and the prototypical graph contrastive learning loss, the overall loss function for CL-KCRec is presented as follows:

$$\mathcal{L} = \mathcal{L}_{bpr} + \beta * \mathcal{L}_{cl} \qquad (18)$$

## 5 EXPERIMENTS

In this section, we evaluate our CL-KCRec in comparison to the baselines. We also analyze the impact of key modules and the robustness of the model. Our experiments are designed primarily to address the following research questions:

- **RQ1**: How does CL-KCRec compare to the baselines?
- **RQ2**: Is it beneficial for the key modules to boost the performance of knowledge concept recommendation in MOOCs?
- **RQ3**: Is our CL-KCRec also proficient in showing strong generalization and robustness on other recommendation tasks and datasets?
- **RQ4**: How do hyperparameters impact model performance?

## 5.1 Experimental Settings

**Datasets**. We conducted experiments on the real-world dataset MOOCCube[32], which is constructed from actual student learning behavior data on the *XuetangX*[1] platform. Specially, we select student behaviors that were recorded between January 1st, 2018 and June 30th, 2019 as the dataset used in this work (named as MOOCCube1819). Table 1 presents the data statistics. Moreover, we used March 31st, 2019, as the dividing point, with the earlier part as the training set and the latter part as the test set. Each positive instance in the test set is paired with 99 randomly sampled negative instances, and the output of prediction score is calculated based on these 100 samples (1 positive and 99 negatives)[9].

**Evaluation Metrics**. We adopt three widely used evaluation metrics to evaluate the recommendation performance: *HR@K* (Hit Ratio of top-K items), *NDCG@K* (Normalized Discounted Cumulative Gain) and *MRR* (Mean Reciprocal Rank)[8]. Specifically, we set $K$ to 5, 10, and 20.

[1]https://www.xuetangx.com/

**Table 1: Statistics of the MOOCCube1819 dataset**

| Dataset | | MOOCCube1819 | |
|---|---|---|---|
| # Entities | # **U**ser | | 2,204 |
| | # **K**nowlege concept | | 1,522 |
| | # **C**ourse | | 706 |
| | # **V**ideo | | 1,661 |
| | # **T**eacher | | 1,738 |
| # Relationships | # U-K (user·click·knowledge-concept) | | 928,476 |
| | # U-V (user·watch·video) | | 4,142 |
| | # U-C (user·learn·course) | | 25,956 |
| | # V-K (video·include·knowledge-concept) | | 27,610 |
| | # C-K (course·include·knowledge-concept) | | 142,654 |
| | # C-V (course·include·video) | | 4,838 |
| | # C-T (course·taught-by·teacher) | | 4,364 |

### 5.1.1 Baseline Methods.

To evaluate the performance of our CL-KCRec, we consider several baselines as follows:

- FISM[12]: This is an item-to-item collaborative filtering approach that generates recommendation.
- NeuMF[10]: It uses a multi-layer perceptron to determine the probability of recommending a knowledge concept.
- NAIS[9]: It is a collaborative filtering approach which employs an attention mechanism.
- metapath2vec[4]: This is a classical heterogeneous representation method by random walk and skip-gram.
- HIN2vec[6]: This is a model which can learn latent vectors of nodes and meta-paths simultaneously in HIN.
- HERec[24]: This is a approach to HIN-based recommendation that utilizes HIN embedding with meta-paths.
- ACKRec[8]: This is an end-to-end approach designed for knowledge concept recommendation in MOOCs.
- HetGNN[35]: It learns heterogeneous node embeddings by aggregating type-based node features and neighboring node.
- MHCN[33] It uses a multi-channel hypergraph convolutional network to consider global relationships.
- KGCL[31]: This is a contrastive learning framework for KG-enhanced recommendation.
- CoNR[15]: It learns both node and relation representations by a two-step attention mechanism and relation encoder.
- HGCL[1]: It utilizes heterogeneous relational semantics with contrastive self-supervised learning for recommendation.

### 5.1.2 Hyperparameter Settings.

For a fair comparison, CL-KCRec is optimized with Adam for parameter learning. In the model implementation, the batch size is searched from {1024, 2048, 4096, 8192}. The initial dimension size of node $d_0$ in MHIN is searched from the range of {16, 32, 64, 128}. The learning rate is searched from {2e-2, 3e-2, 3.5e-2, 5e-2}. For each baseline, all other hyperparameters are set the same as the suggestions from the original settings in their papers. Other hyperparameters are set as follows. The number of the basis vectors for representing explicit relations is tuned from the range of {5, 10, 15, 20}; the number of hops $n$ for implicit relation representation is tuned from the range of {2, 3, 4, 5}; the number of clusters is searched from {5, 10, 15, 20, 40, 50}; the mini-batch size

**Table 2: Performance comparison of baselines on the MOOC-Cube1819 dataset**

|  | H@5 | H@10 | H@20 | N@5 | N@10 | N@20 | MRR |
|---|---|---|---|---|---|---|---|
| NeuMF | 0.2470 | 0.4843 | 0.6757 | 0.2238 | 0.2391 | 0.2755 | 0.2054 |
| FISM | 0.2595 | 0.4892 | 0.6886 | 0.2382 | 0.2456 | 0.3047 | 0.2121 |
| NAIS | 0.2756 | 0.5022 | 0.7011 | 0.2591 | 0.2647 | 0.3231 | 0.2436 |
| HERec | 0.3957 | 0.5875 | 0.7660 | 0.3051 | 0.3599 | 0.4008 | 0.2901 |
| ACKRec | 0.4566 | 0.6287 | 0.8159 | 0.3570 | 0.4114 | 0.4548 | 0.3490 |
| MHCN | 0.4421 | 0.6293 | 0.8205 | 0.3568 | 0.4045 | 0.4542 | 0.3413 |
| KGCL | 0.4584 | 0.6428 | 0.8288 | 0.3597 | 0.4129 | 0.4610 | 0.3454 |
| HGCL | 0.4657 | 0.6572 | 0.8364 | 0.3624 | 0.4201 | 0.4705 | 0.3598 |
| metapath2vec | 0.3190 | 0.5314 | 0.7252 | 0.2736 | 0.2912 | 0.3462 | 0.2673 |
| HIN2vec | 0.3370 | 0.5551 | 0.7449 | 0.2944 | 0.3191 | 0.3789 | 0.2880 |
| HetGNN | 0.4208 | 0.6022 | 0.7927 | 0.3313 | 0.3985 | 0.4241 | 0.3244 |
| CoNR | 0.4593 | 0.6462 | 0.8315 | 0.3778 | 0.4188 | 0.4668 | 0.3687 |
| **CL-KCRec** | **0.5136** | **0.6751** | **0.8417** | **0.4163** | **0.4313** | **0.5163** | **0.4032** |

**Table 3: Ablation study on key components of CL-KCREC.**

|  |  | H@5 | H@10 | H@20 | N@5 | N@10 | N@20 | MRR |
|---|---|---|---|---|---|---|---|---|
| | *w/-er* | 0.4762 | 0.6217 | 0.7996 | 0.3648 | 0.3697 | 0.4859 | 0.3738 |
| | *w/-ir* | 0.4987 | 0.6431 | 0.8255 | 0.3965 | 0.3991 | 0.5088 | 0.3862 |
| | *w/o-cl* | 0.5002 | 0.6598 | 0.8267 | 0.3987 | 0.4046 | 0.5100 | 0.3937 |
| *w/o-att* | *w/-⊕* | 0.5108 | 0.6689 | 0.8392 | 0.4147 | 0.4278 | 0.5109 | 0.3989 |
| | *w/-+* | 0.5086 | 0.6610 | 0.8321 | 0.4078 | 0.4193 | 0.5067 | 0.3944 |
| **CL-KCRec** | | **0.5136** | **0.6751** | **0.8417** | **0.4163** | **0.4313** | **0.5136** | **0.4032** |

is set to 8; the temperature $\tau$ is tuned in {0.3, 0.5, 0.6}; the coefficient $\lambda$ of L2 regularization is set to 1e-4, and the coefficient $\beta$ of the combined contrastive loss is tuned in {0.2, 0.25, 0.3, 0.35, 0.55}.

## 5.2 Performance Comparison (RQ1)

Table 2 displays the performance of all the baselines on the MOOC-Cube1819 for knowledge concept recommendation tasks. We summarize the following observations and conclusions. Our CL-KCRec consistently outperforms state-of-the-art benchmarks, demonstrating significant improvements in performance metrics. We attribute this performance improvement primarily to: (1) Our CL-KCRec represents not only the explicit relations but also the implicit ones to capture users' interests more accurately. (2) The contrastive learning with prototypical graph can enhance the representation, capturing both explicit and implicit relational knowledge, which can guide the propagation of students' preferences. (3) The dual-head attention mechanism can significantly fuse the enhanced representations to balance their contributions for knowledge concept recommendation task.

## 5.3 Ablation Study (RQ2)

We conduct ablation study to validate the significance and benefits of each module. *w/-er* and *w/-ir*: We consider either the explicit or implicit relations in the MHIN. *w/o-cl*: In this variant, we do not include the contrastive learning module. *w/o-att*: In this variant, we do not include the dual-head attention mechanism. Instead, we directly utilize them for recommendation after processing through either vector concatenation *w/-⊕* or addition *w/-+* methods.

The performance of our CL-KCRec and the compared variants are presented in Table 3. The performance of CL-KCRec is superior

**Table 4: Statistics of Yelp and Douban Movie Datasets**

| Datesets | Yelp | | Douban Movie | |
|---|---|---|---|---|
| # Entities | # User | 16,018 | # User | 13,224 |
| | # Business | 14,192 | # Movie | 12,498 |
| | # Compliment | 11 | # Group | 2,747 |
| | # City | 47 | # Director | 2,358 |
| | # Category | 511 | # Actor | 6,251 |
| | | | # Type | 38 |
| # Relationships | # User-Business | 194,552 | # User-Movie | 1,007,399 |
| | # User-User | 156,090 | # User-User | 4,085 |
| | # User-Compliment | 76,555 | # User-Group | 568,783 |
| | # Business-City | 13,970 | # Movie-Director | 11,245 |
| | # Business-Category | 39,927 | # Movie-Actor | 33,051 |
| | | | # Movie-Type | 27,443 |

to both *w/-ex* and *w/-im*, reflecting that relying solely on either explicit or implicit relation is inadequate. *w/o-cl* performs worse than CL-KCRec, which demonstrates that by enhancing representations through contrastive learning, it can address the issue caused by the quantitative disparity between explicit and implicit relations, where the inherent relational knowledge struggles to effectively guide the propagation of interests among students. The performance of *w/o-att* (with *w/-⊕* and *w/-+*) are inferior compared to CL-KCRec, which further implies the necessity of the dual-head attention mechanism for recommendation.

## 5.4 Generalization and Robustness Analysis (RQ3)

This work primarily focuses on the knowledge concept recommendation in MOOCs. To validate the generalization of our CL-KCRec for other recommendation tasks and its robustness across different datasets, we conducted further experimental verification. We adopt two more widely used datasets from different domains, consisting of Yelp Datasets[2] from business domain and Douban Movie Datasets[3] from movie domain, as shown in Table 4. And Table 5 displays the performance of all compared methods on these two datasets for item recommendation. Compared with the baselines, our CL-KCRec still demonstrates significant performance advantages, proving its excellent generalization capability and robustness across other domain recommendation tasks and datasets.

## 5.5 Hyperparameter Analysis (RQ4)

We further perform parameter sensitivity analysis to show the impact of key parameters. The results are presented in Figure 5. Based on the results, we make the following conclusions. *Relation Basis Vectors*. The number of basis vectors that used for representing the explicit relations $\mathcal{B}$ is selected from 5 to 20. We observe that the performance of the model initially increases and then stabilizes. The value of $\mathcal{B}$ at which it stabilizes varies across different datasets: for the MOOCCube1819 dataset, $\mathcal{B} = 15$; while for the DMovie and Yelp datasets, $\mathcal{B} = 10$. *Implicit Relation Hops*. The number of the implicit relation hops $t$ is selected from 2 to 5. It can be seen that our model achieves optimal performance and remains stable

---

[2]https://www.yelp.com/dataset
[3]http://movie.douban.com

**Table 5: Performance comparison of all methods on different datasets**

| Datasets | Yelp | | | | | | | Douban Movie | | | | | | |
|---|---|---|---|---|---|---|---|---|---|---|---|---|---|---|
| | H@5 | H@10 | H@20 | N@5 | N@10 | N@20 | MRR | H@5 | H@10 | H@20 | N@5 | N@10 | N@20 | MRR |
| HERec | 0.6264 | 0.7854 | 0.8663 | 0.4362 | 0.4825 | 0.5321 | 0.4301 | 0.4451 | 0.6172 | 0.7512 | 0.3248 | 0.3781 | 0.4188 | 0.3004 |
| ACKRec | 0.6522 | 0.8046 | 0.8957 | 0.4631 | 0.5137 | 0.5426 | 0.4789 | 0.4829 | 0.6489 | 0.7685 | 0.3825 | 0.4096 | 0.4531 | 0.3521 |
| MHCN | 0.6547 | 0.8314 | 0.8864 | 0.5232 | 0.5823 | 0.6154 | 0.4776 | 0.4767 | 0.6457 | 0.7763 | 0.3966 | 0.4172 | 0.4526 | 0.3648 |
| KGCL | 0.6875 | 0.8547 | 0.9004 | 0.5589 | 0.6219 | 0.6410 | 0.5051 | 0.5134 | 0.6974 | 0.7885 | 0.4034 | 0.4326 | 0.4877 | 0.3764 |
| HGCL | 0.6983 | 0.8656 | 0.9217 | 0.5751 | 0.6382 | 0.6438 | 0.5398 | 0.5441 | 0.7205 | 0.8078 | 0.4299 | 0.4615 | 0.4973 | 0.3922 |
| **CL-KCRec** | **0.7078** | **0.8742** | **0.9254** | **0.5842** | **0.6455** | **0.6602** | **0.5427** | **0.5631** | **0.7259** | **0.8103** | **0.4335** | **0.4697** | **0.5011** | **0.4035** |

when the hops of implicit relations are 4 and 5. This further indicates that in MHIN, deeper implicit relations include more complex semantics, thereby enhancing recommendation performance. *The number of prototypes.* The number of clusters is chosen from 5 to 50. We observe that based on the scale of nodes across different datasets, the optimal number of prototypes varies. To achieve the best recommendation performances, the optimal number of prototypes for the MOOCCube1819 dataset is 10, while for the DMovie and Yelp datasets, it's 40.

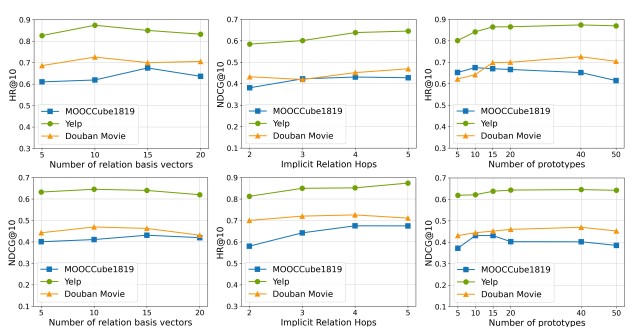

**Figure 5: Hyperparameter study of the CL-KCRec.**

### 5.6 Case Study

In this part, we conduct one case to demonstrate the effectiveness of our proposed framework **CL-KCRec**. We randomly selected user:10843058 and obtained three recommended list without IRs, with IRs and with CL-KCRec, respectively. As shown in Figure 6, based on the click histories and their enrollment in course:30240184, there might be implicit connections, such as shared interests or similar knowledge levels, between user:10843058 and user:10196388. When user:10843058's actual next-clicked knowledge concept *Resistor* (highlighted in dark blue), is observed, it is recommended at the 3rd rank (highlighted in dark green) in list(b). This represents a significant improvement from its 8th rank in list(a). Futhermore, related knowledge concepts such as *logic symbol*, *coil*, and *current* are recommended more frequently. This demonstrates that implicit relations play a crucial role. In list(c), when using our CL-KCRec, the *Resistor* is ranked at the 2nd position, and related concepts are recommended more prominently.

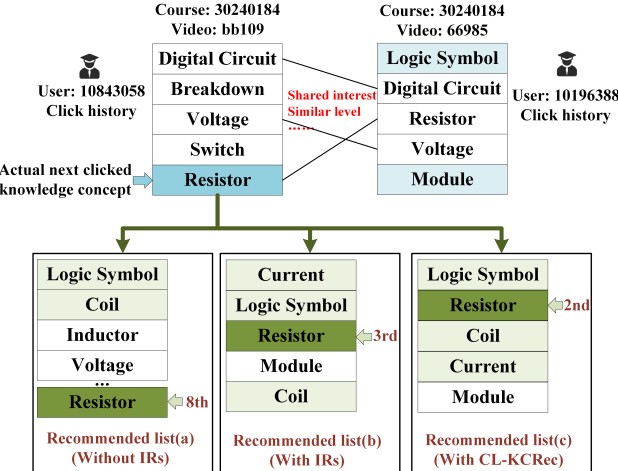

**Figure 6: The case study of CL-KCRec.**

## 6 CONCLUSION

In this work, our proposed CL-KCRec framework explores a novel approach based on contrastive learning for the knowledge concept recommendation in MOOCs. It can automatically represent implicit relations within the MOOCs heterogeneous information network. Furthermore, the contrastive learning with prototypical graph can address the challenge of effectively guiding the propagation of students' preferences, which is caused by the quantitative disparity between explicit and implicit relations. The dual-head attention mechanism can address the imbalanced contributions of these relations for knowledge concept recommendation in MOOCs. This work emphasizes the significant role of implicit relations in knowledge concept recommendation, contributing to the enhancement of the quality of personalized learning services in MOOCs. Extensive experiments on multiple real-world datasets have demonstrated that CL-KCRec outperforms various state-of-the-art methods.

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
