# OpenReview forum: "Modeling Balanced Explicit and Implicit Relations with Contrastive Learning for Knowledge Concept Recommendation in MOOCs"
_ACM.org/TheWebConf/2024/Conference — TheWebConf24_

### Official Review · Reviewer_vZ13 · 2023-11-03

**Novelty:** 3
**Technical Quality:** 4

**Review:**

This work aims to handle knowledge concept recommendation, where the recommender systems provide knowledge concepts of interest for users in Massive Open Online Courses (MOOCs). The authors argued that the existing methods of knowledge concept recommendation primarily rely on the explicit interactions between users and knowledge concepts, while ignoring implicit relations generated within the users’ learning activities. To this end, they proposed novel framework called CL-KCRec which can represent and balance the explicit and implicit relations for knowledge concept recommendation in MOOCs. Specifically, a heterogeneous information network is first constructed to encode various data of MOOCs platforms. Afterwards, a relation-updated graph convolutional network (GCN) and stacked multi-channel graph neural network (GNN) is devised to represent the explicit and implicit relations in the MHIN, respectively. Moreover, a contrastive learning with prototypical graph is designed to enhance the representations of both relations. In addition, a dual-head attention mechanism is further presented for balanced relation fusion. Extensive experiments are conducted to demonstrate the effectiveness of the proposed method.

Strength:

1.	Modeling both explicit and implicit relations to improve Knowledge Concept Recommendation is rational and promising.

2.	The research topic, i.e., knowledge concept recommendation, is interesting.

3.	Overall, the paper is well writing and easy to follow.


Weakness:

1.	The motivation seems not clear. To name a few, some crucial concepts are not explained in detail when being presented, like explicit relations and implicit relations. Line 106-139, the difficulties for capturing implicit relations are not clear. The authors just put forward their view-of-points without providing strong supports.

2.	The rationales of some techniques are weak. To name a few, why did the author choose different GNNs to model explicit and implicit relations? Why did the proposed model apply contrastive learning?

3.	The main claim of the paper is a little confusing and inappropriate. In this work, it seems that the authors define one-hop relation as explicit relations while multi-hop relations are viewed as implicit relations. However, as we all know, the graph neural network, like heterogeneous graph in this case, can learn multi-hop relations among nodes by conducting information aggregation. Therefore, it is inappropriate to claim that the existing works do not consider implicit relations. This also reduces the novelty of the work.

4.	Some experiment details are missing. For instance, Line 207, what is the ‘interactive data of a user’. The model rely on user interactive data to predict concepts of interest, while it is not clearly elaborated. Besides, Line 324, 'initial representation of nodes by BERT with their auxiliary information.' what is auxiliary information? how to represent them?

5.	Some model design is confusing. The authors obtain implicit relations by aggregating various kinds of explicit relations as shown in Eq(7-9). As stated in 3, GNN can learn implicit relations (multi-hop relations). Thus, why did the author design such a complex method to learn implicit relations? The rationale of the design choice should be detailed.

6.	There is a lack of supports for some points. For instance, ‘the quantity of explicit relations is obviously fewer than that of implicit relations’. How the calculate the quantity of relations? And, as stated in the paper, the implicit relations are derived from various kinds of explicit relations. Thus, Why are there more implicit relationships than explicit relationships?

7.	Overall, the experiments are extensive. However, there only one dataset used to evaluate model performance, which could not strongly demonstrate the effectiveness of the proposed method. It is suggested that other datasets can be added to examine the performance of the proposed method.

8.	Some editor: Line 296, 311, 372, 403 and so on, the format of the third level title is incorrect. There is a redundant ‘.’ at the end of the sentence.

**Questions:**

Please see weakness for details

**Reviewer Confidence:**

4: The reviewer is certain that the evaluation is correct and very familiar with the relevant literature

**Scope:**

4: The work is relevant to the Web and to the track, and is of broad interest to the community

---

### Official Review · Reviewer_VUmx · 2023-11-23

**Novelty:** 4
**Technical Quality:** 5

**Review:**

This work proposes a framework, CL-KCRec, for modeling balanced explicit and implicit relations for knowledge concept recommendation in MOOCs. The authors argue that existing recommendation models for MOOCs often fail to capture implicit relations between knowledge concepts, which can lead to suboptimal recommendations. To address this issue, CL-KCRec utilizes a heterogeneous information network (HIN) to represent both explicit and implicit relations between knowledge concepts, and employs contrastive learning to enhance the representation of these relations.

However, many concepts (definitions) given in this paper are the same as existing ones, such as MHIN is just traditional HIN. On the other hand, several concepts, such as prototypical graph, are not given in the paper.

In addition, several important related work, such as “Prototypical Graph Contrastive Learning” in TNNLS 2022, are missing in the paper.

**Questions:**

Does the GCN procedure in “Explicit Relation Learning” also make use of the implicit relations? What is the difference with the “Implicit Relationship Learning”?

**Reviewer Confidence:**

4: The reviewer is certain that the evaluation is correct and very familiar with the relevant literature

**Scope:**

4: The work is relevant to the Web and to the track, and is of broad interest to the community

---

### Official Review · Reviewer_HP7x · 2023-11-27

**Novelty:** 5
**Technical Quality:** 5

**Review:**

Summary:
This paper highlights the importance of balancing explicit user interactions (like clicks and course enrollments) and implicit relations (such as shared interests) on MOOC platforms. The approach involves using a Graph Convolutional Network (GCN) and a stacked multi-channel Graph Neural Network (GNN) for explicit and implicit relation learning, respectively, enhanced by contrastive learning with a prototypical graph. This method has shown to outperform several state-of-the-art models in terms of hit rate, normalized discounted cumulative gain (NDCG), and mean reciprocal rank (MRR).

Strengths:
1. The paper addresses both explicit and implicit user relations, offering a more holistic view of user behavior and preferences in MOOCs.
2. The use of GCN and GNN for different types of relations is an innovative approach that effectively captures the complex dynamics of MOOC platforms.
3. The incorporation of contrastive learning with a prototypical graph significantly improves the representation of user relations, leading to more accurate recommendations.
4. The proposed method demonstrates superior performance over several existing models, indicating its effectiveness and practical applicability.

Weaknesses:
1. The model's effectiveness relies on the quality and comprehensiveness of the data from MOOC platforms, which might vary.
2. It's unclear how well the model would generalize to other online learning platforms or different types of educational content.
3. The complexity of the model might raise concerns about computational efficiency, especially when scaling to large numbers of users and courses.

**Questions:**

1. How do you ensure that the complexity of your model does not hinder its practical deployment, especially for MOOC platforms with limited computational resources?
2. Could you elaborate on how the model handles variations in data quality across different MOOC platforms?

**Reviewer Confidence:**

3: The reviewer is confident but not certain that the evaluation is correct

**Scope:**

4: The work is relevant to the Web and to the track, and is of broad interest to the community

---

### Official Review · Reviewer_4Euf · 2023-11-30

**Novelty:** 5
**Technical Quality:** 5

**Review:**

Pros:
1.The paper provides a novel infomation network that consists of Heterogeneous data from MOOC platform, and this network outperforms state-of-the-art baselines.
2.Implicit relations are added to the network in order to harness more infomation from user-item interaction and user-user relationships. Also, GCN and GNN are used to make explicit and implicit infomation well-selected and merged.
3.This work proposes a contrastive learning with prototypical graph to enhance the representations of both relations, and propose a dualhead attention mechanism for balanced fusion.
4.Abundant experiments are conducted to verify the conponents of the network, which show the significance and benefits of each module.
Cons:
1.Lack of experiment on efficiency study: for the online education platform, inference time is an important factor, so nessesary experiments and analysis need to be added.

**Questions:**

1.How efficient is your proposed framework? Can some experiment be provided?
2.Can the method be extended to some common used datasets , and be conpared with classical CF baselines?

**Reviewer Confidence:**

4: The reviewer is certain that the evaluation is correct and very familiar with the relevant literature

**Scope:**

4: The work is relevant to the Web and to the track, and is of broad interest to the community

---

### Official Review · Reviewer_E6Nv · 2023-12-01

**Novelty:** 5
**Technical Quality:** 5

**Review:**

The proposed system uses both explicit and implicit relationships (e.g., transitive relationships) to train representations for entities in a MOOC -- users, courses, knowledge concepts, videos, etc. Each representation is trained independently. The implicit relationship is computed as a multi-hop product of the explicit relation matrix. Finally, the authors concatenate the two representations and use it for the recommendation problem.


Strong
* Comprehensive experimental analysis by comparing against multiple baselines on open datasets.
* Provides values for all hyper parameters to help with repro.
* Representing any embedding as a combination of basis vectors is a nice way to regularize.

Weak
* Computational cost isn’t mentioned in the experiments. Many of the techniques proposed are hard to scale to a large number of users/items. In fact, the experiments only deal with a few thousand users and courses.
* the paper is pretty hard to read, with too much notation. please consider simplifying.

Details:
* It would be nice to properly define implicit vs. explicit relationships right in the introduction itself.
* Please provide more details on the computational cost.

**Questions:**

Details:
* It would be nice to properly define implicit vs. explicit relationships right in the introduction itself.
* Please provide more details on the computational cost.

**Ethics Review Description:**

no concerns

**Reviewer Confidence:**

2: The reviewer is willing to defend the evaluation, but it is likely that the reviewer did not understand parts of the paper

**Scope:**

3: The work is somewhat relevant to the Web and to the track, and is of narrow interest to a sub-community

---

### Decision · Program_Chairs · 2024-01-22

**Decision:**

Accept

**Comment:**

All reviewers found merits in the submission and they are generally satisfied with it, especially in the novel application of MOOC scenarios. However, they still propose a number of concerns including (1) the efficiency issue (no related experimental results are provided). (2) a more formal definition of implicit and explicit relations. (3) more evidence to show the improvement compared with baselines. I would encourage the authors to revise their manuscript per the reviewers' comments if it can be finally accepted.